# Healthcare resource utilisation and costs associated with a heart failure diagnosis: a retrospective, population-based cohort study in Sweden

Kurt Boman [ID],[1] Krister Lindmark,[2] Jan Stålhammar,[3] Mona Olofsson,[1] Madlaina Costa-Scharplatz,[4] Ana Filipa Fonseca,[5] Stina Johansson,[6] Vincent Heller,[7] Michael Törnblom,[8] Gerhard Wikström[9]

For numbered affiliations see end of article.

**Correspondence to**
Professor Kurt Boman;
kurt.boman@
regionvasterbotten.se

## ABSTRACT

**Objectives** To examine healthcare resource use (HRU) and costs among heart failure (HF) patients using population data from Sweden.

**Design** Retrospective, non-interventional cohort study.

**Setting** Two cohorts were identified from linked national health registers (cohort 1, 2005–2014) and electronic medical records (cohort 2, 2010–2012; primary/secondary care patients from Uppsala and Västerbotten).

**Participants** Patients (aged ≥18 years) with primary or secondary diagnoses of HF (≥2 International Classification of Diseases and Related Health Problems, 10th revision classification) during the identification period of January 2005 to March 2015 were included.

**Outcome measures** HRU across the HF phenotypes was assessed with logistic regression. Costs were estimated based on diagnosis-related group codes and general price lists.

**Results** Total annual costs of secondary care of prevalent HF increased from SEK 6.23 (€0.60) to 8.86 (€0.85) billion between 2005 and 2014. Of 4648 incident patients, HF phenotype was known for 1715: reduced ejection fraction (HFrEF): 64.5%, preserved ejection fraction (HFpEF): 35.5%. Within 1 year of HF diagnosis, the proportion of patients hospitalised was only marginally higher for HFrEF versus HFpEF (all-cause (95% CI): 64.7% (60.8 to 68.4) vs 63.7% (60.8 to 66.5), HR 0.91, p=0.14; cardiovascular disease related (95% CI): 61.1% (57.1 to 64.8) vs 60.9% (58.0 to 63.7), HR 0.93, p=0.28). Frequency of hospitalisations and outpatient visits per patient declined after the first year. All-cause secondary care costs in the first year were SEK 122 758 (€12 890)/ patient/year, with HF-specific care accounting for 69% of the costs. Overall, 10% of the most expensive population (younger; predominantly male; more likely to have comorbidities) incurred ~40% of total secondary care costs.

**Conclusions** HF-associated costs and HRU are high, especially during the first year of diagnosis. This is driven by high hospitalisations rates. Understanding the profile of resource-intensive patients being at younger age, male sex and high Charlson comorbidity index scores at the time of the HF diagnosis is most likely a sign of more severe disease.

## STRENGTHS AND LIMITATIONS OF THIS STUDY

⇒ The study provides comprehensive information from two county regions in Sweden on the increase in number of newly diagnosed heart failure (HF) patients, resource use across the HF phenotypes and total annual costs of secondary care of prevalent HF in Sweden.

⇒ The study provides new insights about the profile of resource-intensive patients at the time of the HF diagnosis, which can support guidance in resource allocation and improve patient management.

⇒ A retrospective identification of patients with confirmed HF despite the requirement for at least two HF diagnoses is challenging.

⇒ Primary and secondary care resource use and costs were available only for the Uppsala region cohort, which limited the direct comparison with the data from the Västerbotten region where only secondary care resources were available.

⇒ Our data reflect the real-world costs associated with a diagnosis of HF, despite an acknowledged limitation in the true validation of the diagnosis due to lack of echocardiogram records for the majority of patients.

## INTRODUCTION

Heart failure (HF) represents a major public health challenge in developed countries and is associated with high healthcare resource consumption[1] with cardiovascular diseases (CVDs) as one of the leading cause of mortality in Sweden.[2] HF places a considerable burden on the healthcare systems,[3 4] and approximately 20% of patients discharged from hospitals after an HF admission were readmitted within 30 days between 2010 and 2013.[5] Inpatient HF care accounts for the largest proportion of healthcare costs,[6] and that the total annual direct cost related to HF

is Swedish krona 2.6 billion, corresponding to approximately 2% of the total Swedish healthcare budget.[7]

An ageing population, increasing number of comorbidities and changed treatment patterns necessitate the examination of healthcare resource use (HRU) and costs. Therefore, our primary aim was to investigate trends in total healthcare costs from 2005 to 2014 among prevalent and incident patients with HF.[8] Patient characteristics in the context of resource use were analysed to determine which individuals incurred the greatest resource and secondary care costs. With the growing importance of HF with preserved ejection fraction (HFpEF), this investigation goes a step further than other Swedish studies[3 4] to present new data comparing resource use and costs among patients according to HF phenotype.

## METHODS
### Study design
This was a retrospective, non-interventional cohort study using longitudinal patient-level data for patients with HF in Sweden.[9] Data were collected from electronic medical records (EMRs) from primary and secondary care and from local echocardiography registries in the counties of Uppsala (2 hospitals and 46 primary care centres (PCCs)) and Västerbotten (3 hospitals and 37 PCCs). EMR data were linked via unique identifiers issued by the National Board of Health and Welfare (NBHW) to data from national health registers, including the National Patient Register (NPR). Data were pseudonymised before the linked database was released to the research group.

### Patients
Patients (aged ≥18 years) with primary or secondary diagnoses of HF during the identification period of January 2005 to March 2015 were included. Analysing data from patients treated in both primary and secondary care by linking NPR data with EMR data presents the challenge of distinct HF diagnosis specificity across the two settings. Two HF diagnoses were required to reduce the uncertainty surrounding HF diagnosis in primary care and increase overall diagnosis specificity. Diagnosis of HF was based on the International Classification of Diseases and Related Health Problems, 10th revision (ICD-10) diagnostic codes of I50, I42.0, I42.6, I42.7, I42.9, I11.0, I13.0 or I13.2 for the primary or secondary diagnoses (online supplemental table S1).

The national cohort comprised prevalent patients with HF registered in the NPR from secondary care between January 2005 and December 2014, while the regional cohort (Uppsala and Västerbotten) included incident patients with HF diagnoses registered in EMRs from either primary or secondary care between January 2010 and March 2015.

The Pygargus Customized eXtraction Program (CXP 3.0) extracted data from EMRs from Uppsala and Västerbotten and subsequently linked to data from the national health registers including the NPR.[10] HF with preserved ejection fraction (HFpEF) and HF with reduced ejection fraction (HFrEF)) was determined based only on data from local echocardiography registries, and for cohort 2, HFpEF was defined as a left ventricular ejection fraction (LVEF) of ≥50% and HFrEF was defined as an LVEF of <50%. This cut-off is based on data constraints; LVEF is recorded categorically in the echocardiography registers, with the categories used differing between the county councils such that 50% was the only option that allowed for a common threshold.

### Data extraction and study timelines
Data were extracted from EMRs based on an observed HF diagnosis during 1994–2015 for the Uppsala County and during 1992–2016 for the Västerbotten County. The analysis period was adjusted to 1 January 2010 owing to inconsistent patterns of HF registration in the Uppsala County EMR systems. The look-back period in the EMR data was from the time the first data were available until 31 December 2009. For NPR data, the look-back period was from 1 January 1997 for inpatient care and 1 January 2001 for outpatient care to 31 December 2004 (online supplemental figure S1) to identify prevalent patients with HF at a national level. This prevalent patient cohort was used to assess trends in total HRU and costs from 2005 to 2014.

Newly diagnosed individuals from Uppsala County and during the analysis period made up the incident HF population. To enable all patients to have at least 3 years of potential follow-up, all analyses were performed on patients diagnosed with incident HF between 2010 and 2012 (online supplemental figure S1).

For the analyses of clinical characteristics, HRU and costs, the index date was defined as the date of first HF diagnosis. However, for the analyses of hospitalisations, the index date was defined as a patient's second HF diagnosis to improve the accuracy of diagnosis and minimise the effects of immortal time bias. Follow-up was defined as the period between the second HF diagnosis and the end of the study, end of EMR collection for those patients who moved to another region, or date of death, whichever came first.

### Variables analysed and statistical analyses
For categorical outcomes a $\chi^2$ test was performed; for comparison of means, a t-test was performed and CIs were calculated at a 95% level. Descriptive statistics assessed baseline clinical characteristics (table 1) on the date of first HF diagnosis, since this was truly the first time that the patients were considered to have HF. Comorbidities, data from primary (EMR) and secondary (EMR and NPR) care stratified according to HF phenotype, were summarised according to the ICD-10 codes (online supplemental table S1). Comorbidities were collected from visits that occurred 0–5 years before the date of the first HF diagnosis. Charlson comorbidity index (CCI) scores during this time, ranging from 0 to greater than

**Table 1** Demographic and clinical characteristics at baseline for incident patients with HF from the Uppsala and Västerbotten counties with a first HF diagnosis in 2010–2012, overall and by HF phenotype

| Characteristic | Overall population (N=5205) | HF phenotype | | |
| --- | --- | --- | --- | --- |
| | | HFpEF (n=652) | HFrEF (n=1167) | Unknown LVEF (n=3386) |
| Mean age at index date (SD), years | 76.8 (12.3) | 74.8 (12.2) | 70.3 (13.0) | 79.4 (11.1) |
| Median age at index date (IQR), years | 79.0 (69.5– 85.7) | 76.8 (68.3–83.6) | 71.2 (63.1–80.3) | 81.5 (73.4–87.2) |
| Sex, n (%) | | | | |
| Women | 2405 (46.2) | 341 (52.3) | 380 (32.6) | 1684 (49.7) |
| Men | 2800 (53.8) | 311 (47.7) | 787 (67.4) | 1702 (50.3) |
| Mean NT-proBNP level (SD), pg/mL | 4909.1 (8202.9) | 4270.8 (7576.7) | 6217.1 (9495.2) | 4543.4 (7730.5) |
| Mean SBP/DBP (SD), mm Hg | 139.7 (25.6)/78.7 (15.3) | 141.8 (27.1)/76.8 (14.6) | 134.9 (24.3)/79.7 (15.3) | 140.9 (25.6)/78.8 (15.3) |
| Mean eGFR (SD), mL/min/1.73 m$^2$ | 51.9 (29.1) | 49.6 (27.9) | 60.6 (30.2) | 44.2 (26.0) |
| Mean CCI*† (SD) | 1.8 (2.2) | 1.8 (2.2) | 1.7 (2.1) | 1.9 (2.2) |
| Common comorbidities and risk factors, n (%)†‡ | | | | |
| Hypertension | 2679 (51.5) | 366 (56.1) | 476 (40.8) | 1837 (54.3) |
| Atrial fibrillation | 1593 (30.6) | 212 (32.5) | 247 (21.2) | 1134 (33.5) |
| IHD (angina or MI) | 1198 (23.0) | 122 (18.7) | 303 (26.0) | 773 (22.8) |
| Diabetes | 952 (18.3) | 137 (21.0) | 197 (16.9) | 618 (18.3) |
| Cancer | 739 (14.2) | 118 (18.1) | 141 (12.1) | 480 (14.2) |
| Dyslipidaemia | 676 (13.0) | 89 (13.7) | 152 (13.0) | 435 (12.8) |
| Cerebrovascular disease | 660 (12.7) | 70 (10.7) | 109 (9.3) | 481 (14.2) |
| Anaemia | 645 (12.4) | 104 (16.0) | 101 (8.7) | 440 (13.0) |

Data are either number (n) and percentage (%) or mean (SD).

*Includes patients with a CCI of zero (ie, no comorbidities).
†Comorbidities and underlying cardiac diseases occurring 0–5 years before the index date. c
‡Comorbidities and underlying cardiac diseases occurring in ≥10% of the overall population.
CCI, Charlson comorbidity index; DBP, diastolic blood pressure; eGFR, estimated glomerular filtration rate; HF, heart failure; HFpEF, HF with preserved ejection fraction; HFrEF, HF with reduced ejection fraction; IHD, ischaemic heart disease; LVEF, left ventricular ejection fraction; MI, myocardial infarction; NT-pro-BNP, N-terminal pro-B-type natriuretic peptide; ;SBP, systolic blood pressure.

10 (higher scores indicate greater comorbidity), were calculated.

All-cause CVD-related hospitalisations were analysed using a subdistribution hazard model, proportion of patients having all-cause CVD-related hospitalisations occurring within 0.5, 1 and 3 years after the second HF diagnosis were estimated by a cumulative incidence function. The time to all-cause and time to CVD-related hospitalisations/readmissions were estimated from the date of the second HF diagnosis using a Cox proportional hazards model, with age group, sex, diagnosis setting (primary care, secondary care, unknown), HF phenotype and N-terminal pro B-type natriuretic peptide (NT-proBNP) level (0–1000, 1001–3000 and >3000 pg/mL) as covariates. Data for the mean and median (IQR)

number of hospitalisations and outpatient visits for all-cause, CVD-related and HF-related events, as well as mean length of stay (LOS), were analysed by HF phenotype and year since HF diagnosis.

Costs were estimated by year since the first HF diagnosis. All costs are presented as SEK 2015 values (based on the Consumer Price Index),[11] with the equivalent value in euros (€), based on the historical exchange rate in January 2015 (1 SEK=€0.105). Analyses of all-cause, CVD-related and HF-related costs associated with secondary care were performed based on diagnosis-related group codes and price lists, as determined by the NBHW.[12] Costs associated with primary care comprised costs of family physician and nurse visits as well as costs of blood tests. Primary care costs were based on general price lists

and were available for Uppsala County only, as were costs related to pharmacotherapy.[13] The total cost of all drug use for incident patients with HF was estimated, as was the aggregated cost of HF-related drugs, based on the 2012 European Society of Cardiology (ESC) guidelines (online supplemental table S2).[14] Logistic regression assessed baseline characteristics of patients who incurred the greatest secondary care costs related to HF, defined as the top 10% most resource-intensive patients.

SAS V.9.3 or higher was used for statistical analysis and data management.

### Patient and public involvement

No patients or members of the public were involved in this study.

## RESULTS

### National prevalent population with HF

The absolute number of prevalent patients with HF in Sweden increased by 1.5-fold between 2005 and 2014 from 89 837 to 133 220 (overall population increased from 7 113 513 to 7 762 073), with a corresponding increase in age-adjusted prevalence from 1.30% to 1.72%.

### Increase in total costs of prevalent patients with HF over time

The total number of hospitalisations, hospital days, emergency room (ER) visits and outpatient visits in secondary care increased from 2005 to 2014 by between 1.2-fold and 2.2-fold (online supplemental figure S2). The number of all-cause hospitalisations increased from 141 941 to 181 374 (difference: 39 433; 28% increase). The number of hospital days increased from 998 512 to 1 165 310 (difference: 166 798; 17% increase), indicating a decreased mean LOS of 4.23 days per hospitalisation. ER visits increased from 14 272 to 31 037 (difference: 16 765; 117% increase). Outpatient visits in secondary care increased from 262 771 to 497 230 (difference: 234 459; 89% increase).

The total annual costs associated with the secondary care of prevalent patients with HF increased by 1.4-fold from SEK 6.23 billion (€0.60 billion) in 2005 to SEK 8.86 billion (€0.85 billion) in 2014, mainly driven by a corresponding increase in the overall resource use in inpatient care (52%) (online supplemental figure S3A). On an average, inpatient and outpatient costs accounted for 84% and 16% of the total costs, respectively. On an average, 81% of the total costs of secondary care were attributed to CVD-related costs; 64% of these were attributed to HF-related costs (online supplemental figure S3B). HF-related costs accounted for 52% of the total costs.

### Regional incident population with HF

Data from 104 562 patients were identified from the Uppsala and Västerbotten counties; among these, 8702 (8.3%) patients had a defined HF diagnosis. Information was available for 4648 incident patients with HF between 2010 and 2012 (online supplemental figure S1). HF phenotype was characterised for 1715 patients (36.9%); 1106 (64.5%) patients had HFrEF and 609 (35.5%) had HFpEF (online supplemental figure S1).

### Patient demographics and clinical characteristics

Table 1 presents the baseline clinical characteristics of patients, overall and by HF phenotype. Mean (median, IQR) age of the patients was 76.8 (79.0, 69.5–86.7) years, more than one-quarter (27.8%) were aged 85 years or more, and 53.8% were male. Most first HF diagnoses occurred in secondary care (75.9%) rather than in primary care (24.1%) (table 2).

### Hospitalisations

#### All-cause and CVD-related hospitalisations

The cumulative incidence function (95% CI) of the proportion of patients with all-cause hospitalisations within the first year of HF diagnosis was 64.7% (60.8 to 68.4) for patients with HFrEF and 63.7% (60.8 to 66.5) for patients with HFpEF; the corresponding subdistribution HR over the follow-up period for HFrEF compared with HFpEF was 0.91 (95% CI 0.81 to 1.03; p=0.14), cardiovascular disease related (95% CI): 61.1% (57.1 to 64.8) vs 60.9% (58.0 to 63.7), HR 0.93, p=0.28). (online supplemental table S3). The cumulative incidence function of the proportion of patients with all-cause hospitalisations was similar for patients who had their first HF diagnosis in secondary versus primary care; the corresponding subdistribution HR for the follow-up period for secondary vs primary care was 0.93 (95% CI 0.86 to 1.01; p<0.07) (online supplemental table S3). Higher NT-proBNP levels were associated with a greater risk of all-cause hospitalisations (online supplemental table S3), also seen for the proportion of patients with CVD-related hospitalisations by HF phenotype, diagnosis setting, NT-proBNP level and age (table 2).

#### Number of hospitalisations and outpatient visits and LOS

Mean and median number of all-cause, CVD-related and HF-related hospitalisations and outpatient visits per patient decreased from the first year of HF diagnosis and stabilised between years 2 and 3 (figure 1). Similar trends in hospitalisations and outpatient visits compared with those observed in the overall population were observed for patients with HFrEF and HFpEF, except that in patients with HFpEF, the mean number of CVD-related outpatient visits increased from the first year after HF diagnosis (figure 1). The mean (median, IQR) LOS for all-cause, CVD-related and HF-related hospitalisations was 17.5 (8, 2–23), 16.5 (8, 2–22) and 12.8 (6, 0–16) days in the first year after HF diagnosis, declining at 3 years after diagnosis to 7.6 (0, 0–8), 7.0 (0, 0–7) and 4.2 (0, 0–2) days, respectively. Patients with HFrEF had a shorter LOS for all-cause, CVD-related and HF-related events than those with HFpEF (data not shown).

**Table 2** Subdistribution hazard model for the proportion of incident patients with HF from the Uppsala and Västerbotten counties with a second HF diagnosis in 2010–2012 who had CVD-related hospitalisations within 3 years and cumulative incidence function estimates for the proportion of patients hospitalised within 0.5, 1 and 3 years after index date

| Characteristic | N=4648 | Cumulative incidence function of CVD-related hospitalisations, % (95% CI) | | | Subdistribution HR (95% CI) | P value |
| | | 0.5 years | 1 year | 3 years | | |
|---|---|---|---|---|---|---|
| Age group, years | | | | | | |
| 18–54 (reference) | 275 | 46.9 (40.9 to 52.7) | 52.0 (45.9 to 57.7) | 62.5 (55.9 to 68.3) | 1 | |
| 55–64 | 457 | 47.0 (42.4 to 51.5) | 56.5 (51.8 to 60.9) | 71.8 (66.7 to 76.3) | 1.15 (0.95 to 1.41) | 0.16 |
| 65–74 | 1051 | 47.3 (44.2 to 50.3) | 57.5 (54.4 to 60.4) | 72.5 (69.2 to 75.4) | 1.21 (1.01 to 1.45) | 0.0383 |
| 75–84 | 1592 | 44.6 (42.1 to 47.0) | 55.3 (52.9 to 57.7) | 75.6 (72.8 to 78.0) | 1.21 (1.02 to 1.45) | 0.0305 |
| ≥85 | 1273 | 41.9 (39.2 to 44.6) | 52.2 (49.4 to 54.9) | 70.7 (67.8 to 73.4) | 1.13 (0.94 to 1.35) | 0.19 |
| Sex | | | | | | |
| Female (reference) | 2143 | 43.8 (41.7 to 45.9) | 52.9 (50.8 to 55.0) | 71.7 (69.4 to 73.9) | 1 | |
| Male | 2505 | 45.8 (43.8 to 47.7) | 56.5 (54.6 to 58.4) | 73.0 (70.9 to 74.9) | 1.06 (0.98 to 1.14) | 0.12 |
| Diagnosis setting | | | | | | |
| Primary care (reference) | 1118 | 41.8 (38.9 to 44.6) | 53.3 (50.3 to 56.2) | 77.1 (73.8 to 80.0) | 1 | |
| Secondary care | 3528 | 45.8 (44.2 to 47.4) | 55.3 (53.7 to 57.0) | 71.0 (69.3 to 72.7) | 0.94 (0.86 to 1.02) | 0.12 |
| Unknown | 2 | N/A | N/A | N/A | 1.99 (1.32 to 3.00) | 0.0011 |
| HF phenotype | | | | | | |
| HFrEF (reference) | 609 | 53.2 (49.2 to 57.1) | 61.1 (57.1 to 64.8) | 77.9 (73.7 to 81.5) | 1 | |
| HFpEF | 1106 | 52.4 (49.5 to 55.3) | 60.9 (58.0 to 63.7) | 73.8 (70.8 to 76.6) | 0.93 (0.83 to 1.06) | 0.28 |
| Unknown LVEF | 2933 | 40.3 (38.5 to 42.0) | 51.3 (49.5 to 53.1) | 70.9 (68.9 to 72.8) | 0.79 (0.71 to 0.88) | <0.0001 |
| NT-proBNP level, pg/mL | | | | | | |
| 0–1000 (reference) | 912 | 40.4 (37.2 to 43.5) | 51.8 (48.5 to 54.9) | 72.4 (68.8 to 75.7) | 1 | |
| 1001–3000 | 1283 | 45.4 (42.7 to 48.1) | 56.6 (53.8 to 59.2) | 76.1 (73.1 to 78.7) | 1.10 (0.99 to 1.21) | 0.07 |
| >3000 | 1601 | 52.5 (50.0 to 54.9) | 61.3 (58.8 to 63.6) | 74.3 (71.7 to 76.7) | 1.19 (1.08 to 1.32) | 0.0006 |
| Missing | 852 | 34.5 (31.3 to 37.7) | 43.5 (40.2 to 46.8) | 63.2 (59.4 to 66.9) | 0.83 (0.74 to 0.93) | 0.0017 |

The Fine and Grays subdistribution hazard model was used to obtain the subdistribution HRs. Cumulative incidence function was used to estimate percentage of patients with CVD-related hospitalisations within 0.5, 1 and 3 years after index date.
CVD, cardiovascular disease; HF, heart failure; HFpEF, HF with preserved ejection fraction; HFrEF, HF with reduced ejection fraction; LVEF, left ventricular ejection fraction; NT-proBNP, N-terminal pro-B-type natriuretic peptide.

### Healthcare costs per patient for the first year after the index date

#### Secondary healthcare (overall, inpatient and outpatient) costs

Total and inpatient all-cause costs decreased by more than half after the first year of HF diagnosis (figure 2). A reduction in all-cause outpatient costs was also noted at the second year after HF diagnosis, relative to the first year. Similar trends were seen for CVD-related and HF-related costs. In the first year after HF diagnosis, total all-cause secondary care costs were SEK 122 758 (€12 890) per patient per year, which declined to SEK 53 220 (€5588) per patient per year by year 2 after diagnosis. The corresponding total CVD-related and HF-related costs were SEK 110 268 (€11 578) and SEK 84 956 (€8920) per patient per year in the first year, which decreased to SEK 43 590 (€4577) and 26 038 (€2734) per patient per year by year 2, respectively. Therefore, HF-specific inpatient and outpatient costs accounted for 69% of the total

costs in the first year after HF diagnosis, declining to 49% during the second year after diagnosis (and 46% during the third year after diagnosis). Inpatient care constituted more than 90% of the total secondary healthcare costs per patient at 1 year after diagnosis for all-cause (SEK 112 432 (€11 805)), CVD-related (SEK 106 101 (€11 141)) and HF-related (SEK 82 897 (€8704)) events. By comparison, the mean total primary care cost for the Uppsala County cohort only was SEK 10 347 (€1086) per patient per year (online supplemental table S4, information).

### Drivers of high resource use and costs

Of the total patient population, 10% (n=465) incurred approximately 40% of the total secondary healthcare costs. Compared with 'other' (non-resource intensive) patients, those in the resource-intensive population were likely to have a lower mean age, belong to the male sex, have a mean NT-proBNP level above 3000 pg/mL, and a

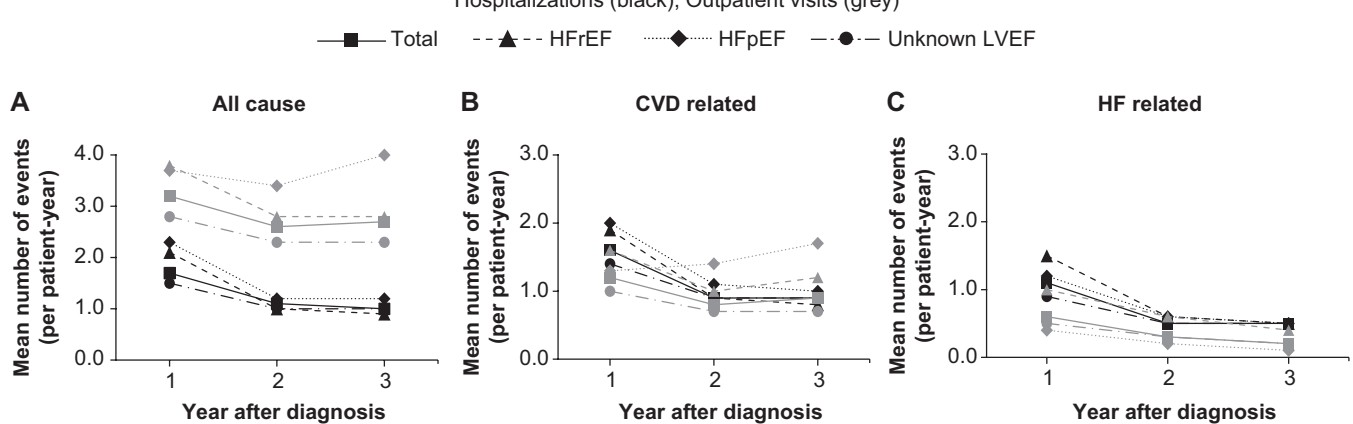

**Figure 1** Mean number of hospitalisations (lines in black) and outpatient visits (lines in grey) for incident patients with HF during 2010–2012 by year after HF diagnosis. Number of patients 1, 2 and 3 years after diagnosis: 4648, 2769 and 1309 total patients; 1106, 667 and 323 patients with HFrEF; 609, 353 and 156 patients with HFpEF; 2933, 1749 and 830 with unknown LVEF, respectively. CVD, cardiovascular disease; HF, heart failure; HFpEF, HF with preserved ejection fraction; HFrEF, HF with reduced ejection fraction; LVEF, left ventricular ejection fraction.

number of comorbidities; had a known LVEF; and had not received either an ACE inhibitor or angiotensin receptor blocker (table 3).

## DISCUSSION

This study found that (i) the absolute number of patients with HF in Sweden continues to increase; (ii) the total cost of secondary care of prevalent patients with HF increased and was primarily driven by costs associated with inpatient care; (iii) patients with newly diagnosed HF were at a high risk of frequent hospitalisations during the first year after diagnosis, especially in those diagnosed in secondary vs primary care; (iv) HFpEF was associated with greater resource use compared with HFrEF or an unknown LVEF; (v) during the first year, hospitalisations (inpatient care) constituted more than 90% of the total

secondary healthcare costs, with 69% of all secondary healthcare costs being attributable to HF-specific care and (iv) approximately 10% of the patients were highly resource-intensive.

In HF management, hospitalisations are a key challenge because they are highly resource intensive and costly.[3 4 15–17] Not only are patients with HF frequently hospitalised, but duration of their admissions are also often lengthy, particularly in Europe compared with the USA.[15 16] In the current study, the mean hospital LOS during the first year after HF diagnosis was more than 16 days for patients admitted for all-cause or CVD-related events and 12.8 days for patients admitted for HF-related events.

The clinical implication of our data is to identify factors responsible for the high costs of HF within the first year

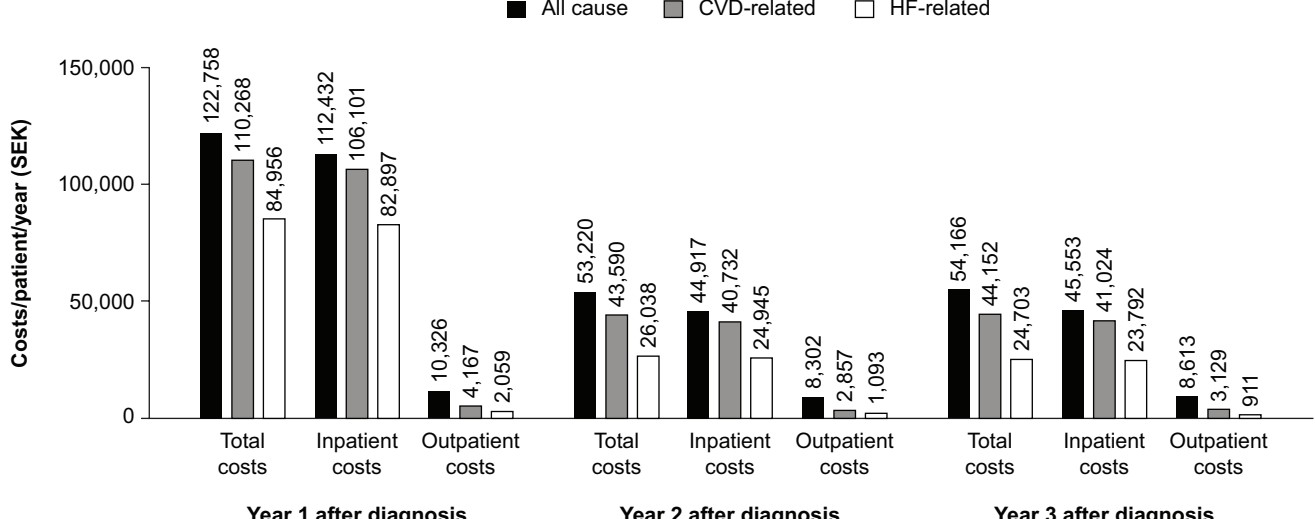

**Figure 2** All-cause, CVD-related and HF-related total, inpatient and outpatient costs up to 3 years after diagnosis of HF (incident patients with HF, 2010–2012). Number of patients: year 1, n=4648; year 2, n=2769; year 3, n=13095. All costs are given in SEK 2015 values (1 SEK=€0.105). CVD, cardiovascular disease; HF, heart failure; SEK, Swedish krona.

**Table 3** Logistic regression to determine characteristics at HF diagnosis of the 10% of incident patients with HF incurring the greatest secondary healthcare costs during the first year after index date

| Characteristics at HF diagnosis | N=5205 n (%) | Unadjusted model | | Adjusted model | |
|---|---|---|---|---|---|
| | | OR | 95% CI | OR | 95% CI |
| Age at index date, years | | | | | |
| 18–54 | 297 (5.7%) | Reference | | Reference | |
| 55–64 | 511 (9.8%) | 0.88 | 0.60 to 1.29 | 0.83 | 0.56 to 1.24 |
| 65–74 | 1155 (22.2%) | 0.89 | 0.64 to 1.24 | 0.82 | 0.57 to 1.18 |
| 75–84 | 1796 (34.5%) | 0.40 | 0.28 to 0.57 | 0.39 | 0.27 to 0.57 |
| ≥85 | 1446 (27.8%) | 0.18 | 0.12 to 0.27 | 0.21 | 0.13 to 0.33 |
| Sex | | | | | |
| Male | 2800 (53.8%) | Reference | | Reference | |
| Female | 2405 (46.2%) | 0.73 | 0.61 to 0.88 | 0.98 | 0.80 to 1.20 |
| LVEF | | | | | |
| HFrEF | 1167 (22.4%) | Reference | | Reference | |
| HFpEF | 652 (12.5%) | 1.01 | 0.79 to 1.30 | 1.05 | 0.80 to 1.39 |
| Unknown | 3386 (65.1%) | 0.29 | 0.24 to 0.36 | 0.41 | 0.33 to 0.52 |
| NT-proBNP, ng/L | | | | | |
| 0–1000 | 1075 (20.7%) | Reference | | Reference | |
| 1001–3000 | 1433 (27.5%) | 1.17 | 0.88 to 1.56 | 1.48 | 1.09 to 2.01 |
| >3001 | 1730 (33.2%) | 1.77 | 1.36 to 2.30 | 1.93 | 1.45 to 2.57 |
| Missing | 967 (18.6%) | 1.17 | 0.85 to 1.60 | 1.44 | 1.03 to 2.02 |
| ACEI or ARB treatment | | | | | |
| ACEI | 2481 (47.7%) | Reference | | Reference | |
| ACEI and ARB | 58 (1.1%) | 1.81 | 0.88 to 3.74 | 1.61 | 0.75 to 3.45 |
| ARB | 964 (18.5%) | 1.02 | 0.79 to 1.31 | 1.01 | 0.77 to 1.33 |
| Neither ACEI nor ARB | 1702 (32.7%) | 1.26 | 1.03 to 1.55 | 1.63 | 1.31 to 2.03 |
| Hypertension | | | | | |
| No | 2526 (48.5%) | Reference | | Reference | |
| Yes | 2679 (51.5%) | 1.11 | 0.93 to 1.33 | 1.06 | 0.84 to 1.33 |
| Atrial fibrillation | | | | | |
| No | 3612 (69.4%) | Reference | | Reference | |
| Yes | 1593 (30.61) | 0.75 | 0.61 to 0.92 | 0.77 | 0.61 to 0.97 |
| Angina or MI | | | | | |
| No | 4007 (76.9%) | Reference | | Reference | |
| Yes | 1198 (23.02%) | 1.45 | 1.19 to 1.78 | 1.27 | 1.00 to 1.60 |
| Diabetes | | | | | |
| No | 4253 (81.71%) | Reference | | Reference | |
| Yes | 952 (18.29%) | 1.48 | 1.19 to 1.83 | 1.19 | 0.93 to 1.53 |
| Cancer | | | | | |
| No | 4466 (85.80%) | Reference | | Reference | |
| Yes | 739 (14.20%) | 1.19 | 0.93 to 1.52 | 1.17 | 0.89 to 1.53 |
| Cerebrovascular disease | | | | | |
| No | 4545 (87.32%) | Reference | | Reference | |
| Yes | 660 (12.68%) | 1.08 | 0.83 to 1.41 | 1.27 | 0.95 to 1.71 |
| Dyslipidaemia | | | | | |
| No | 4529 (87.01%) | Reference | | Reference | |

Continued

| Characteristics at HF diagnosis | N=5205 n (%) | Unadjusted model | | Adjusted model | |
|---|---|---|---|---|---|
| | | OR | 95% CI | OR | 95% CI |
| Yes | 676 (12.99%) | 1.60 | 1.26 to 2.03 | 1.10 | 0.83 to 1.47 |
| Anaemia | | | | | |
| No | 4560 (87.61%) | Reference | | Reference | |
| Yes | 645 (12.39%) | 1.73 | 1.36 to 2.20 | 1.63 | 1.24 to 2.14 |
| COPD | | | | | |
| No | 4801 (92.24%) | Reference | | Reference | |
| Yes | 404 (7.76%) | 1.27 | 0.93 to 1.74 | 1.13 | 0.80 to 1.58 |
| Aortic insufficiency/regurgitation | | | | | |
| No | 4814 (92.49%) | Reference | | Reference | |
| Yes | 391 (7.51%) | 3.21 | 2.49 to 4.13 | 2.54 | 1.92 to 3.37 |
| Dementia | | | | | |
| No | 4883 (93.81%) | Reference | | Reference | |
| Yes | 322 (6.19%) | 0.33 | 0.19 to 0.60 | 0.53 | 0.29 to 0.99 |
| Chronic kidney disease | | | | | |
| No | 4987 (95.81%) | Reference | | Reference | |
| Yes | 218 (4.19%) | 2.47 | 1.76 to 3.48 | 1.44 | 0.97 to 2.14 |
| Depression | | | | | |
| No | 5015 (96.35%) | Reference | | Reference | |
| Yes | 190 (3.65%) | 1.38 | 0.90 to 2.13 | 1.21 | 0.75 to 1.93 |
| Mitral insufficiency/regurgitation | | | | | |
| No | 5030 (96.64%) | Reference | | Reference | |
| Yes | 175 (3.36%) | 3.32 | 2.34 to 4.72 | 2.58 | 1.74 to 3.81 |
| Peripheral artery disease | | | | | |
| No | 5086 (97.71%) | Reference | | Reference | |
| Yes | 119 (2.29%) | 2.47 | 1.57 to 3.87 | 1.60 | 0.97 to 2.65 |

ACEI, ACE inhibitor; ARB, angiotensin receptor blocker; COPD, chronic obstructive pulmonary disease; HF, heart failure; HFpEF, heart failure with preserved ejection fraction; HFrEF, heart failure with reduced ejection fraction; LVEF, left ventricular ejection fraction; MI, myocardial infarction; NT-proBNP, N-terminal pro-B-type natriuretic peptide.

of diagnosis. Such factors could include patients being discharged from hospital too early to ensure full evaluation, patients not being initiated on evidence-based therapy or devices, delayed or incomplete follow-up in patient management programmes, and loss of continuity and/or ineffective HF follow-ups.

An important observation concerned the risk of hospitalisation/readmission over the follow-up period, which was significantly greater for patients receiving their first HF diagnosis in secondary versus primary care. This might be the result of patients presenting with more severe HF and/or more severe chronic HF and patients with a diagnosis of acute HF. Accordingly, the total costs associated with primary care, available for the Uppsala County only, were SEK 10 347 (€1086) per patient and were much lower than the secondary care costs in the first year after diagnosis. These costs generally remained stable over the first 4 years of follow-up and increased slightly at year

5, which could be a result of the increasing age of the population.

Patients with HFpEF were also at a higher risk of all-cause and CVD-related hospitalisations and required more outpatient visits at 3 years than those with HFrEF, potentially because patients with HFpEF tended to be older and/or have a higher comorbidity burden. Importantly, a higher proportion of deaths in patients with HFpEF than in those with HFrEF may have a non-CVD cause.[18][19] Mortality data presented in Lindmark et al[20] show that the HR for the 1-year all-cause mortality rate was significantly lower for HFrEF than for HFpEF (0.77, 95% CI 0.62 to 0.96, p=0.0159).

We found a decline in resource use observed after the first year of diagnosis and substantial reductions in healthcare costs, with total all-cause costs per patient decreasing by more than half in the second year. This decrease reflects patient-level, longitudinal data trends

and, in fact, from a total population/healthcare system perspective, the opposite was true with an increase in costs over time because of an increasing prevalence of HF and a shift in patient profile towards older and sicker patients.[20] In our incident population, the high financial costs during the first year after diagnosis may be partly explained by survivor bias, assuming that patients alive in later years have less severe HF symptoms and require less cost-intensive treatment, such as the need for re-hospitalisations and longer LOS. Other possible explanations include the use of both more intensive investigations to define the aetiology of HF during the first year and expensive medicines, devices and/or surgical treatments during the initial phase of treatment. The costs reported herein are generally similar to those reported in another Swedish EMR-based and register-based study that analysed data from 2006.[4]

Similar to other economic studies in HF,[3 21 22] the inpatient care accounted for most of the total secondary care costs (~90%) and showed the greatest proportional decline in costs from the peak at 1 year after diagnosis, consistent with the reduction seen in the number of hospitalisations. Similar trends were observed for CVD-related and HF-related costs per patient. Data for the 10% of patients who accumulated the highest healthcare costs showed that patients with several different comorbidities were associated with a higher resource use. The most resource-intensive patients tended to be male with high NT-proBNP levels. Patients with high/increasing biomarker levels tend to be followed up closely by specialists because of the potential for adverse outcomes.[23]

A limitation of the current study was the challenge of retrospectively identifying patients with confirmed HF. The use of two, rather than one, HF diagnoses was expected to increase the certainty surrounding the HF diagnosis, although it could potentially have led to exclusion of more recent or mild cases of HF. Second limitation concerned the definition of subgroups based on the HF phenotype. The limited number of echocardiographs required for the verification of the HF diagnosis was a major drawback, meaning that a large proportion of patients had missing information for both LVEF and structural and functional pathologies. This may have introduced some level of bias because it could be assumed that the echocardiography was performed only in more severe cases of HF, meaning that the two HF phenotype subgroups were not wholly representative of the total cohort. Nevertheless, our data reflect the real-world costs associated with a diagnosis of HF, irrespective of the true validation of the diagnosis. Moreover, the threshold for defining HFrEF was an LVEF of ≤50%, which is higher than the recommendations from the ESC of <40%[24]; therefore, a larger number of patients would have been categorised as having HFrEF. Finally, there were missing data on doctor visits for the Västerbotten County cohort because of technical issues relating to the Västerbotten EMR database; thus, primary care resource use and costs were available only for the Uppsala County cohort.

## CONCLUSIONS

Care for patients with HF represents an increasing financial burden on the Swedish healthcare system. Hospitalisations/readmissions represent a key resource and cost burden regardless of HF phenotype, particularly during the first year after diagnosis, with total costs of HF attributed primarily to those associated with inpatient care. Furthermore, characteristics of patients incurring the greatest healthcare costs included younger age, male sex, and high CCI scores and NT-proBNP levels at the time of the HF diagnosis. Understanding the profile of patients with HF who are likely to incur the greatest healthcare costs is invaluable to guide resource allocation.

**Author affiliations**
[1]Research Unit, Medicine, Department of Public Health and Clinical Medicine, Umea University, Skellefteå, Sweden
[2]Department of Public Health and Clinical Medicine and Heart Centre, Umea University, Umea, Sweden
[3]Department of Public Health and Caring Sciences, Uppsala University, Uppsala, Sweden
[4]Medical Affairs/RWE, Novartis Sweden AB, Stockholm, Sweden
[5]Novartis Pharma AG, Basel, Switzerland
[6]IQVIA, Stockholm, Sweden
[7]IQVIA, Solna, Sweden
[8]Real-World & Analytics Solutions, IQVIA Solutions Sweden AB, Solna, Sweden
[9]Institute of Medical Sciences, Uppsala University Hospital, Uppsala, Sweden

**Acknowledgements** The authors thank Japinder Kaur of Novartis and Sharon Smalley and Carly Sellick of Pharma Genesis, London, for providing medical writing assistance with this manuscript.

**Contributors** KB and GW were involved in the study and manuscript conceptualisation, interpretation of results and development of the first draft of the manuscript. KL contributed to protocol development, study and manuscript conceptualisation, results interpretation, refinement of analysis requirements and writing of the manuscript. SJ, VH, MT performed statistical analyses. JS and MO contributed to protocol development, study conceptualisation, interpretation of results and supported the development of the manuscript. MC-S and AFF contributed to manuscript conceptualisation, interpretation of results, refinement of analysis requirements and supported the development of the manuscript. All authors reviewed the manuscript and approved the final draft of the manuscript for submission.

**Funding** This research was funded by Novartis Pharma AG, Basel, Switzerland.

**Competing interests** KB, MO and JS received reimbursement from Novartis via IQVIA for performing the study. KB and MO have also received lecture grants from Novartis. KL received lecture grants and consultant fees from Novartis. GW has no conflicts of interest to declare; however, Uppsala University received research funding from Novartis for conducting this study. MC-S is an employee of Novartis Sweden AB, Sweden, and AFF is an employee of Novartis Pharma AG, Basel, Switzerland. Data extraction and analysis were conducted by SJ, VH and MT are employees of IQVIA, Sweden, which was commissioned to conduct the study on behalf of Novartis Pharma AG, Basel, Switzerland, and have ongoing consulting and research relationships with Novartis Pharma AG.

**Patient consent for publication** Not applicable.

**Ethics approval** Ethical approval was obtained from the regional Ethical Review Board in Uppsala, Sweden (2015-045) before data extraction. No informed consent was required for this retrospective, pseudonymised study.

**Provenance and peer review** Not commissioned; externally peer reviewed.

**Data availability statement** All data relevant to the study are included in the article or uploaded as supplementary information. Major findings from the study will be published in scientific manuscripts only. The data will not be made available in any other format in order to preserve the privacy of the patients in compliance with local laws and regulation.

**ORCID iD**
Kurt Boman http://orcid.org/0000-0002-0350-2132

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
