## [Reviewer comments · BMJ Open]

ARTICLE DETAILS

TITLE (PROVISIONAL)	Healthcare resource utilization and costs associated with a heart failure diagnosis: a retrospective, population-based cohort study in Sweden.
AUTHORS	Boman, Kurt; Lindmark, Krister; Stålhammar, Jan; Olofsson, Mona; Costa-Scharplatz, Madlaina; Fonseca, Ana Filipa; Johansson, Stina; Heller, Vincent; Törnblom, Michael; Wikström, Gerhard

VERSION 1 – REVIEW

REVIEWER	Al-Mohammad, Abdallah sheffield teaching hospitals, Cardiology
REVIEW RETURNED	05-Jun-2021

GENERAL COMMENTS	I would like to congratulate the authors on completing this important study of the health care expenditure on heart failure, both prevalent and incident in Sweden over two periods. The authors have been supported by Novartis within its projects of Serelaxin and LCZ696. The authors have defined heart failure through a criterion that the diagnosis should be mentioned in the data-base twice. While this ascertains the diagnosis is correctly made, but could exclude a number of patients with heart failure. The latter is a limitation acknowledged by the authors. The authors made a number of interesting observations with regards to the cost of care for patients with heart failure. Their findings with regards to the different phenotypes may not necessarily be as robust as one would hope as these findings were governed by who had their phenotype recorded which is subject to chance rather than being necessarily a true reflection of the reality. The manuscript is well written. However, I have a few issues that I would like to invite the authors to address: A. There is a minor issue in the paper, and that is the variable mention of the end of the data collection period which is frequently mentioned as 2014 and then as March 2015. It would be nice if the authors were to clarify from the outset that for the prevalent HF data the end was December 2014, while for the incident HF data the end was March 2015. Similarly, they should also say that the onset of the data was 2005 for the prevalent HF and was 2010 for the incident HF.
---

	B. In the discussion section, page 16, lines 24 and 26 state: [The comparatively lower HF-related costs suggest that patients accrue significant costs related to conditions inherent to HF.] This is not a very clear sentence and frankly does not make sense particularly given the data presented before it. I would be grateful to the authors for revising this. C. The use of the two HF diagnoses was recognised by the authors as a potential limitation that could have led to exclusion of mild or more recent diagnoses of HF. It would be nice if the authors were to explain within the discussion the advantage of using the two diagnoses as a method of ascertaining the diagnosis in a retrospective study. Professor Abdallah Al-Mohammad
--	---

REVIEWER	Farre, Nuria Hospital del Mar
REVIEW RETURNED	20-Jun-2021

GENERAL COMMENTS	Abstract: What does CCI stand for? I assume it stands for Charlson comorbidity index, but it is not defined in the abstract. Some of the information given in the conclusion is not mentioned in the results section. Introduction: The last paragraph is confusing. What is the aim of the study? It should be clearly stated. Methods:  - The description of the cohorts analyzed is complicated to understand. - In Figure S1, where do the “A total of 5,205 analyzable patients with 3-year follow-up data were identified” come from? It is a higher number of patients than the incident cohort 2. - What diseases were included in the CVD hospitalization group? - Considering the low number of patients with known ejection fraction and (maybe) NTproBNP, I’m not sure whether it is appropriate to include these variables in the Cox analysis. Results:  - In Figure S3 the main cost is attributed to outpatient care whereas the text says the contrary. Is there a mistake in the Figure? - What was considered to be a HF related cost? - Was NTproBNP measured in all patients? How many missing are there? When was it measured, during the hospitalization or when the patient was stable? NT-proBNP does not follow a normal distribution. Median and interquartile range should be used instead. - In general, all the results given make reference to the incident HF cohort (Cohort 2), which is the one that assesses healthcare resource use. Why is Cohort 1 included in this study? - I think the results from Cohort 1 could be very interesting, especially analyzing changes in epidemiology over time. I may merit a different article or at least to include the Figures in the main manuscript and not the Supplement.
--

VERSION 1 – AUTHOR RESPONSE

Reviewer-1 comments for the author

1. There is a minor issue in the paper, and that is the variable mention of the end of the data collection period which is frequently mentioned as 2014 and then as March 2015. It would be nice if the authors were to clarify from the outset that for the prevalent HF data the end was December 2014, while for the incident HF data the end was March 2015. Similarly, they should also say that the onset of the data was 2005 for the prevalent HF and was 2010 for the incident HF.

Response: The 2nd paragraph in patients section of methods (page 5) covers the requested information:

“The national cohort comprised prevalent patients with HF registered in the NPR from secondary care between January 2005 and December 2014, while the regional cohort (Uppsala and Västerbotten) included incident patients with HF diagnoses registered in EMRs from either primary or secondary care between January 2010 and March 2015.”

2. In the discussion section, page 16, lines 24 and 26 state: [The comparatively lower HF-related costs suggest that patients accrue significant costs related to conditions inherent to HF.] This is not a very clear sentence and frankly does not make sense particularly given the data presented before it. I would be grateful to the authors for revising this.

Response: Statement removed from the main document.

3. The use of the two HF diagnoses was recognised by the authors as a potential limitation that could have led to exclusion of mild or more recent diagnoses of HF. It would be nice if the authors were to explain within the discussion the advantage of using the two diagnoses as a method of ascertaining the diagnosis in a retrospective study.

Response: Based on previous studies, the validity of HF diagnoses at the hospitals is high (ESC Heart Failure 2020; 7: 37–46). However when it comes to primary care, the validity of HF diagnoses is much lower. In addition, in the EMR system in the region it was not specified whether HF was the main diagnoses for the condition or an additional diagnoses. Hence, using the criteria of two diagnosis increased the validity of HF diagnoses but it included a bias of potentially excluded accurate mild HF patients with just one diagnosis. Statement edited in the main document (Main document-marked copy: Page no. 14, line no. 22-24).

Reviewer-2 comments for the author

1. **Abstract:** What does CCI stand for? I assume it stands for Charlson comorbidity index, but it is not defined in the abstract. Some of the information given in the conclusion is not mentioned in the results section.

Response: CCI defined, conclusion modified (Main document-marked copy: Page no. 3, line no. 5-7).

2. **Introduction:** The last paragraph is confusing. What is the aim of the study? It should be clearly stated.

Response: Statement modified (Main document-marked copy: Page no. 4, line no. 14-18).

3. **Methods:**

a) The description of the cohorts analyzed is complicated to understand. - In Figure S1, where do the “A total of 5,205 analyzable patients with 3-year follow-up data were identified” come from? It is a higher number of patients than the incident cohort 2.

Response: Thank you for identifying the error, we apologize for it. There was a typo in the description of figure S1 which has been corrected now.

“A total of 4,648 analyzable patients with 3-year follow-up data were identified between 2010 and 2012. Of the 1,715 patients for whom LVEF data were available, 64,5% had HFREF and 35,5% had HFpEF. The incident HF population comprised those with no HF diagnosis in the look-back periods.”

b) What diseases were included in the CVD hospitalization group?

Response: ICD-10 Codes to define CVD-related care: atrial fibrillation (I48), cerebrovascular disease (I60-I69), hypertension (I10-I13, I15), ischemic heart disease (I20-I25), peripheral artery disease (I73.9), stroke (I61-I64).

c) Considering the low number of patients with known ejection fraction and (maybe) NTproBNP, I'm not sure whether it is appropriate to include these variables in the Cox analysis.

Response: Over 80% of patients measured NTproBNT; however, only 63% has no LVEF value. Despite acknowledging that the sample size is not large, we believe that the number of patients with HFrEF (n=609) and HFpEF (n=1106) are considered sufficient from statistical standpoint to be included in the Cox analysis.

4. Results:

a) In Figure S3 the main cost is attributed to outpatient care whereas the text says the contrary. Is there a mistake in the Figure?

Response: Thank you for identifying the error, we apologize for it. Figure S3 has been corrected now.

b) What was considered to be a HF related cost?

Response: All costs for patients with HF diagnoses include: all inpatient and outpatient visits at secondary care, primary care visits with primary care physicians as well as with nurses, all blood tests, costs for comorbidities based on diagnosis-related group codes and all treatments. This is covered in methods section (Main document-marked copy: Page no.8, line no. 5-9).

c) Was NTproBNP measured in all patients? How many missing are there? When was it measured, during the hospitalization or when the patient was stable? NT-proBNP does not follow a normal distribution. Median and interquartile range should be used instead.

Response: NTproBNT was measured in over 80% of the patients. This means in 3796 patients out of the 4648 and only 852 patients were missing NTproBNT measurements. Patients underwent NT-proBNP testing in the period starting 6 months before and ending 6 months after the first HF diagnosis. NTproBNT is included in the local guidelines as part of the normal HF diagnosis work-up process. To be consistent with the previously published abstracts and manuscripts, mean data for NTproBNT was reported.

d) In general, all the results given make reference to the incident HF cohort (Cohort 2), which is the one that assesses healthcare resource use. Why is Cohort 1 included in this study?

Response: Cohort 1 refers to data from the Administrative National Health registries, where all HF patients are included, whereas cohort 2 is a more granular "sub-cohort" including also clinical data from the EMR system.

We agree that the focus of the manuscript is on cohort 2 including comprehensive clinical and administrative data from two specific region of Sweden. In contrast, cohort 1 includes every single HF patients and hence, it allows us therefore to put the results of cohort 2 into the national perspective. This is particularly important when discussing resource use:

We can nicely follow the resource use of individual naive patients in cohort 2 over the years after the diagnoses. But we think that cohort 1 provides great additional value as the results of cohort 2 can then be put into perspective with the total HF costs of the prevalent population in Sweden over time. E.g. The total annual costs associated with the secondary care of prevalent patients with HF in Sweden increased by 1.4-fold from SEK 6.23 billion (€0.60 billion) in 2005 to SEK 8.86 billion (€0.85 billion) in 2014, mainly driven by a corresponding increase in the overall resource use in inpatient care. (Main document-marked copy: Page no.9, line no. 10-13).

e) I think the results form Cohort 1 could be very interesting, especially analyzing changes in epidemiology over time. It may merit a different article or at least to include the Figures in the main manuscript and not the Supplement.

Response: We have considered this but saw a great value to include the results of cohort 1 in this manuscript (as mentioned in comment above). A separate manuscript on the epidemiology data (Clinical Epidemiology 2019;11 231–244) has already been published.

VERSION 2 – REVIEW

REVIEWER	Al-Mohammad, Abdallah sheffield teaching hospitals, Cardiology
REVIEW RETURNED	21-Aug-2021

GENERAL COMMENTS	The revision addressed the issues raised before and I am happy with the manuscript
--

REVIEWER	Farre, Nuria Hospital del Mar
REVIEW RETURNED	09-Aug-2021

GENERAL COMMENTS	The authors have aswered to all my comments.
--